# A Two-Stage Multi-Modal MRI Framework for Lifespan Brain Age Prediction

**Dingyi Zhang**[*1]                                          DINGYI.ZHANG@EMORY.EDU

**Ruiying Liu**[*2]                                                RLIU60@EMORY.EDU

**Yun Wang**[1,2]                                            YUN.WANG2@EMORY.EDU

[1] *Department of Computer Science, Emory University, Atlanta, GA, USA*

[2] *Department of Biomedical Informatics, Emory University, Atlanta, GA, USA*

## Abstract

The accurate quantification of brain age from MRI has emerged as an important biomarker of brain health. However, existing approaches are often restricted to narrow age ranges and single-modality MRI data, limiting their capacity to capture the coordinated macro- and microstructural changes that unfold across the human lifespan. To address these limitations, we developed a multi-modal brain age framework to characterize the integrated evolution of brain morphology and white matter organization. Our model adopts a two-stage architecture, where modalities are processed independently and integrated via late fusion in both stages: first to classify each subject into one of six developmental stages, and then to estimate age within the predicted stage. This design enables a unified and lifespan-spanning assessment of brain maturity across diverse developmental periods.

**Keywords:** Brain Age Prediction, Multi-Modal MRI, Lifespan Brain Development.

## 1. Introduction

Brain age prediction aims to estimate the biological age of the brain from MRI scans, with the gap between predicted and chronological age serving as an indicator of neuroanatomical health (Kumari and Sundarrajan, 2024). With the rapid development of deep learning, a growing number of methods have been proposed for this task, demonstrating promising performance on various neuroimaging datasets (Baecker et al., 2021). However, existing methods still face three key limitations: a focus on restricted age ranges rather than the full lifespan, reliance on single-modality data that fails to capture complementary macro- and microstructural information, and limited robustness to incomplete multimodal inputs in real-world settings.

To address these limitations, we propose a two-stage multi-modal framework for lifespan brain age prediction. Our main contributions are as follows: (1) we present the first brain age prediction framework spanning the complete human lifespan, from fetal to elderly stages, covering the full spectrum of brain development and aging; (2) we adopt a late fusion strategy that integrates T1-weighted (T1w), T2-weighted (T2w), and fractional anisotropy (FA) modalities without requiring multi-modal training, naturally handling missing modalities at inference; (3) we design a two-stage pipeline combining coarse-grained age stage classification and fine-grained brain age regression, with a Mixture of Experts (MoE) (Shazeer et al., 2017) architecture to handle diverse age-specific and modality-specific patterns.

---

[*] Contributed equally

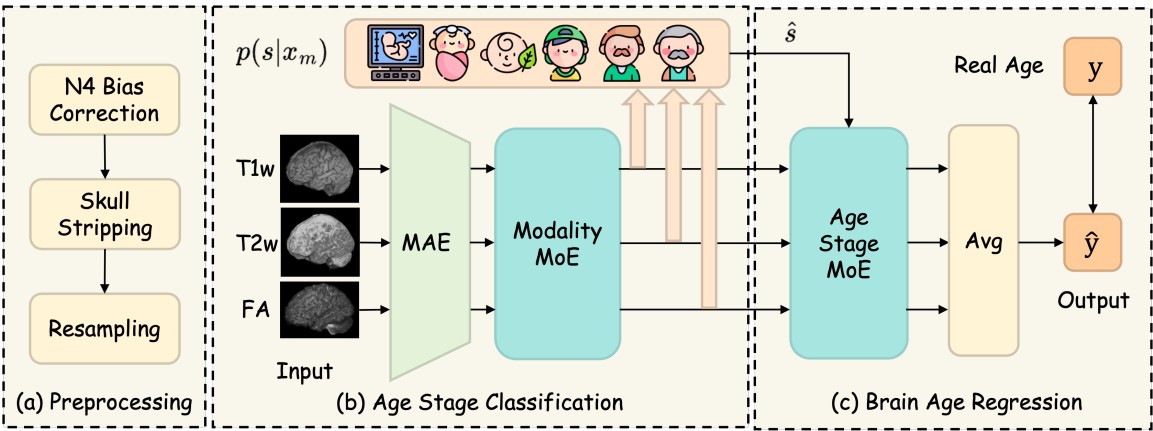

Figure 1: Overview of the proposed two-stage framework for lifespan brain age prediction. Stage 1 predicts $\hat{s}$, and Stage 2 estimates $\hat{y}$ via stage-conditioned regression.

## 2. Method

### 2.1. Problem Formulation

Given a set of MRI scans $\{x_m\}_{m \in \mathcal{M}}$ from available modalities $\mathcal{M} \subseteq \{\text{T1w, T2w, FA}\}$ preprocessed via N4 bias correction, skull stripping (Hoopes et al., 2022), and 1 mm isotropic resampling, our goal is to predict the brain age $\hat{y} \in \mathbb{R}$ of a subject. To unify age representation across the full lifespan, fetal age $w$ (gestational weeks) is mapped to $y = w - 40$, yielding a continuous age axis from prenatal to elderly stages.

### 2.2. Framework

**Age Stage Classification.** In the first stage, we partition the human lifespan into six age stages: fetal (gestational weeks 20–40), neonatal (postnatal weeks 0–12), infant (months 3–24), child (years 2–17), adult (years 18–65), and elderly (years > 65). For each available modality $m$, a shared MAE (He et al., 2022) backbone encodes $x_m$ into a latent representation, which is then processed by a Modality MoE with hard routing, where each expert specializes in a specific MRI modality. This produces a probability distribution $p(s \mid x_m)$ over age stages, where $s$ denotes the age stage. The predicted stage is obtained by aggregating all predictions:

$$\hat{s} = \arg\max_s \sum_m p(s \mid x_m).$$

**Brain Age Regression.** In the second stage, a regression network estimates brain age conditioned on $\hat{s}$ from first stage, enabling fine-grained prediction within each stage. We introduce an Age Stage MoE, where each expert specializes in a specific lifespan stage to capture diverse macro- and microstructural patterns. The shared MAE backbone and Modality MoE are reused and fine-tuned to leverage representations learned during the first stage. Modalities are processed independently, and their predictions are aggregated via late fusion at inference to obtain $\hat{y}$, supporting missing modalities without retraining.

Table 1: Brain age estimation results (MAE / STD in years↓). Best values are **bold**.

| Method | Fetal | Neonatal | Infant | Child | Adult | Elderly |
|---|---|---|---|---|---|---|
| *In-domain* | | | | | | |
| SFCN | 0.87 / 0.10 | 0.62 / 0.04 | 0.45 / 0.30 | **0.96** / 1.24 | **2.90** / 3.93 | 6.54 / **6.96** |
| ORDER | 0.21 / 0.07 | 0.05 / 0.05 | 0.32 / 0.21 | 1.28 / **1.15** | 3.82 / **3.82** | 8.92 / 8.14 |
| TSAN | 0.14 / 0.40 | 0.12 / 0.16 | 0.65 / 1.24 | 1.97 / 2.45 | 3.71 / 4.83 | 6.30 / 8.36 |
| Ours-S | **0.01** / **0.02** | **0.02** / 0.03 | 0.14 / 0.21 | 1.35 / 1.90 | 4.51 / 6.68 | 6.02 / 7.84 |
| Ours-M | **0.01** / **0.02** | **0.02** / **0.02** | **0.11** / **0.15** | 1.17 / 1.63 | 3.97 / 5.86 | **5.77** / 7.31 |
| *Out-of-domain* | | | | | | |
| SFCN | 0.97 / 0.12 | 0.81 / 0.41 | 0.79 / 0.55 | 5.36 / 4.19 | 29.00 / **8.23** | 42.97 / 11.95 |
| ORDER | 0.25 / 0.07 | 0.16 / 0.25 | 0.42 / 0.14 | 7.56 / **3.78** | 30.07 / 12.45 | 36.73 / 22.63 |
| TSAN | 0.68 / 1.11 | 0.72 / 1.92 | 1.43 / 2.66 | 4.68 / 4.57 | 45.71 / 11.61 | 56.30 / 15.08 |
| Ours-S | **0.04** / **0.07** | 0.67 / 5.06 | 0.16 / 0.14 | 6.26 / 11.72 | **12.73** / 12.43 | **6.14** / **6.89** |
| Ours-M | **0.04** / **0.07** | **0.08** / **0.15** | **0.15** / **0.10** | **3.88** / 7.94 | **12.73** / 12.43 | **6.14** / **6.89** |

## 3. Experiments

**Experimental Setup.** We evaluate our method on nine datasets, which are categorized into two groups: in-domain and out-of-domain. A subset of the in-domain datasets is used for training, while the out-of-domain datasets are used exclusively for testing. Both groups cover the full lifespan, ranging from fetal to elderly stages. Specifically, the in-domain datasets include BCP (Howell et al., 2019), dHCP (Eyre et al., 2021), HCP-A (Bookheimer et al., 2019), HCP-D (Somerville et al., 2018), and HCP-YA (Van Essen et al., 2013), while the out-of-domain datasets consist of ABCD (Casey et al., 2018), ADNI (Jack Jr et al., 2008), FeTA (Payette et al., 2023), and HBCD (Volkow et al., 2024). We compare our approach with several baseline methods, including ORDER (Shah et al., 2024), SFCN (Peng et al., 2021), TSAN (Cheng et al., 2021), as well as a single-modal variant of our model.

**Results.** As shown in Table 1, our multi-modal model (Ours-M) achieves the best overall performance across both in-domain and out-of-domain datasets, demonstrating strong generalization ability. It also consistently outperforms its single-modality variant, highlighting the benefit of integrating multiple MRI modalities. Notably, in early developmental stages (fetal and neonatal), the prediction error is close to zero, indicating that the proposed two-stage framework effectively captures age-specific characteristics in these challenging stages.

## 4. Conclusion

We propose a two-stage multi-modal framework for lifespan brain age prediction from MRI, integrating T1w, T2w, and FA modalities via late fusion to accommodate missing modalities. Experiments on nine datasets spanning fetal to elderly stages demonstrate superior performance and strong generalization, with our method reducing MAE by 10% and 69% in in-domain and out-of-domain settings, respectively, compared to existing baselines. Moreover, multi-modal integration yields further gains over the single-modal variant, with improvements of 8% and 11%, respectively, confirming the importance of multi-modal design.

## Acknowledgments

This work was supported by NIH grants R00HD103912, R01MH133313 (Y.W.).

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

## Appendix A. Dataset Distribution

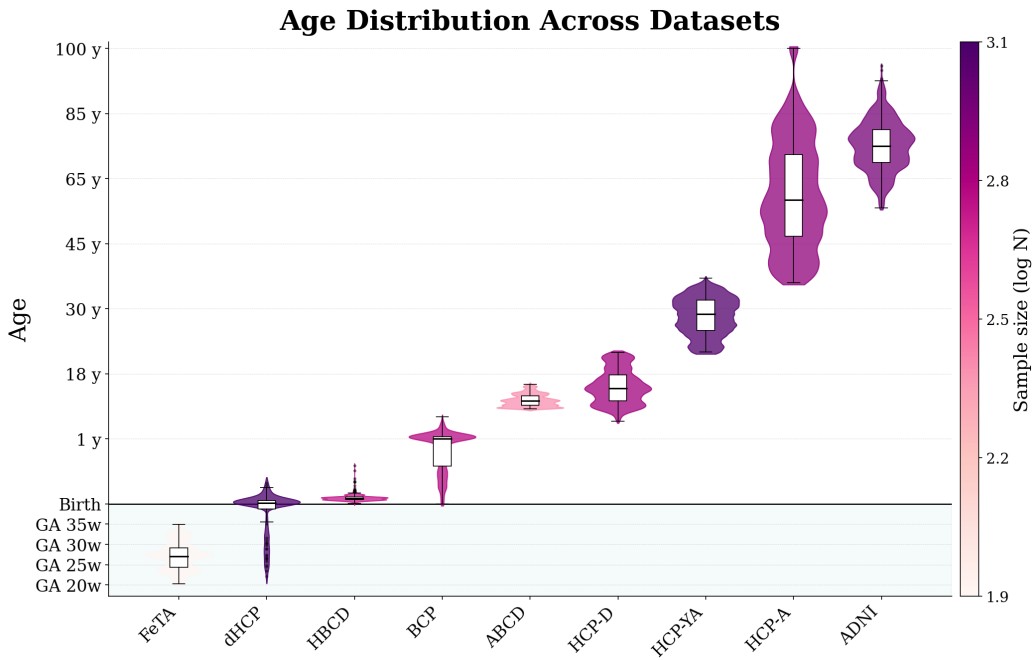

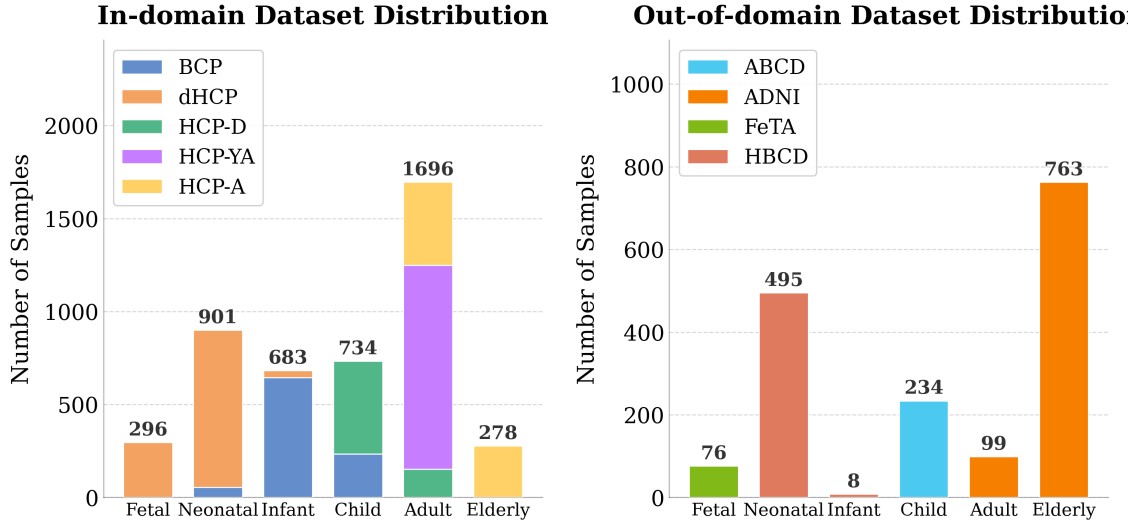

Figure 2: Dataset distribution. Top: age distribution across datasets. Bottom: number of samples per age group and dataset. Each subject-session counts as one sample.

Figure 2 illustrates dataset distribution. The in-domain set comprises five datasets (BCP, dHCP, HCP-D, HCP-YA, and HCP-A) totaling 3804 subjects, 4,588 sessions and 11927 scans, while the out-of-domain set consists of four held-out test datasets (ABCD, ADNI, FeTA, and HBCD) with 1527 subjects, 1,675 sessions and 2251 scans in total.

## Appendix B. Age Stage Classification Results

Figure 3: Confusion matrices of the first stage classifier on the out-of-domain test set.

Figure 3 shows the confusion matrices of the first stage on the out-of-domain test set. Note that the two matrices have different sample counts: Ours-S evaluates each modality independently (2,688 samples), whereas Ours-M aggregates all available modalities per subject into a single prediction (1,675 subjects). Ours-M outperforms Ours-S in overall accuracy (88.18% vs. 80.62%) and eliminates extreme misclassifications: unlike Ours-S, which misclassifies some Neonatal subjects as Adult or Elderly, Ours-M confines all errors to neighboring age groups. Adult accuracy remains low for both models (29%), as out-of-domain adult subjects come exclusively from ADNI (ages 60–65), placing them near the Adult/Elderly boundary.

