# OpenReview forum: "A Two-Stage Multi-Modal MRI Framework for Lifespan Brain Age Prediction"
_MIDL.io/2026/Short_Papers — MIDL 2026 - Short Papers Poster_

### Official Review · Reviewer_TVWx · 2026-04-25
**Interesting paper, which lacks many key details to fully assess what has been done.**

**Rating:** 4
**Confidence:** 5

**Review:**

Interesting work, but there are many details missing that could have been integrated, even with the existing page limits. Results reported for the baselines (e.g., SFCN) appear to be extremely weak for the OOD experiments, which leads to questions regarding the training scheme (overfitting?) and which makes it a bit hard to really assess how good the proposed setup is. I am also struggling a bit with the claims made regarding the first model for the whole lifespan and the late fusion of multiple modalities. Both has been done before and the effectiveness (and need) of the first step (age stage classification) remains unclear.

**Summary:**

Brain age prediction from multi-modal MRI over the life-span using a two stage process (first binning into aging stage, then standard age regression with stage). Uses MAE for representation learning across the two stages and MoE/late fusion to handle missing modalities. Results show improvements over baselines. Ablation wrt single modality network shows some improvements of multi-modal setup. Overall, useful study, technical novelty and design choices remain slightly unclear.

**Strengths:**

- Multimodal brain age prediction with missing modalities is an interesting problem
- Large, diverse dataset used for training and testing
- Several established baseline methods
- Novel two-stage process

**Weaknesses:**

- Motivation for two-stage process and its need remain unclear
- Major claims a bit inflated (see above)
- Results of baselines seem to be very weak
- Evaluation scheme lacks many details (e.g., what modality was used for Ours-S)
- Results only show a slight improvement for multi-modal prediction
- Missingness of modalities not evaluated

**Justification Of Rating:**

Solid paper, with several weaknesses, but I think it passes the bar for a MIDL short paper.

---

### Decision · Program_Chairs · 2026-05-08

Accept (Poster)